# Literature Review and Policy Recommendations for Single-Dose HPV Vaccination Schedule in China: Opportunities and Challenges

**DOI:** 10.3390/vaccines13080786

**Published:** 2025-07-24

**Authors:** Kexin Cao, Yiu-Wing Kam

**Affiliations:** Division of Natural and Applied Science, Duke Kunshan University, No. 8 Duke Avenue, Kunshan 215316, China; kc501@duke.edu

**Keywords:** human papillomaviruses (HPV), cervical cancer, HPV vaccines, single-dose schedule

## Abstract

Cervical cancer remains a significant global public health challenge, with human papillomavirus (HPV) as its primary cause. In response, the World Health Organization (WHO) launched a global strategy to eliminate cervical cancer by 2030 and, in its 2022 position paper, recommended a single-dose vaccination schedule. The objective of this review is to critically examine the current HPV vaccination landscape in China, including vaccination policies, immunization schedules, supply–demand dynamics, and the feasibility of transitioning to a single-dose regimen. By synthesizing recent developments in HPV virology, epidemiology, vaccine types, and immunization strategies, we identify both opportunities and barriers unique to the Chinese context. Results indicate that China primarily adheres to a three-dose vaccination schedule, with an optional two-dose schedule for girls aged 9–14, leaving a notable gap compared to the most recent WHO recommendation. The high prevalence of HPV types 52 and 58 contributes to a distinct regional infection pattern, underscoring the specific need for nine-valent vaccines tailored to China’s epidemiological profile. Despite the growing demand, vaccine supply remains inadequate, with an estimated annual shortfall of more than 15 million doses. This issue is further complicated by strong public preference for the nine-valent vaccine and the relatively high cost of vaccination. Emerging evidence supports the comparable efficacy and durable protection of a single-dose schedule, which could substantially reduce financial and logistical burdens while expanding coverage. This review advocates for the adoption of a simplified single-dose regimen, supported by catch-up strategies for older cohorts and the integration of HPV vaccination into China’s National Immunization Program (NIP). Sustained investment in domestic vaccine development and centralized procurement of imported vaccines may also possibly alleviate supply shortage. These coordinated efforts are critical for strengthening HPV-related disease prevention and accelerating China’s progress toward the WHO’s cervical cancer elimination targets.

## 1. Introduction

Cervical cancer is among the most prevalent cancers affecting women globally, with human papillomavirus (HPV) identified as the primary causative agent. Persistent HPV infection, especially with high-risk types, is responsible for over 95% of cervical cancer cases as well as many HPV-related diseases [1]. Because of its well-established viral origin, cervical cancer is largely preventable, with HPV vaccination serving as a cornerstone of primary prevention and a key strategy in the global effort to eliminate the disease [2]. Recognizing the significant burden of HPV-related diseases and the need for clear, evidence-based vaccination recommendations, the World Health Organization (WHO) has continued to update its HPV immunization guidelines. In 2022, the WHO issued revised recommendations endorsing a single-dose vaccination schedule for certain age groups. This approach aims to simplify implementation, enhance vaccine accessibility, and accelerate progress toward the global goal of cervical cancer elimination [3]. While HPV vaccination efforts are advancing globally, China faces distinct epidemiological and structural challenges that necessitate context-specific strategies.

The focus of this review arises from the WHO’s recent policy update and the ongoing global shift toward reduced-dose HPV vaccination regimens. This topic is particularly timely and relevant for China as the country seeks to expand HPV vaccine coverage and actively respond to the WHO’s cervical cancer elimination initiative. China is now in a critical transition stage in its HPV vaccination policy, with ongoing adjustments to vaccination schedules, coverage strategies, and public health messaging. The frequent evolution of HPV vaccine policies, coupled with regional epidemiological differences and supply constraints, underscores the rationale for an updated and context-specific synthesis of China’s vaccination framework.

Previous local studies have mainly focused on HPV epidemiology or pricing strategies to improve willingness to pay. However, there remains knowledge gaps regarding China’s recent HPV vaccination practices and the feasibility of adopting a single-dose schedule. This review uniquely addresses that gap by providing a comparative analysis of HPV epidemiology and vaccination strategies between the global and China’s latest situation. Special emphasis is placed on the implications of the WHO’s 2022 recommendation supporting single-dose vaccination, assessing its applicability and potential benefits within the China’s context.

By providing a comprehensive and up-to-date synthesis of the HPV vaccination landscape in China, this review identifies existing challenges and opportunities and proposes evidence-based recommendations tailored to China’s national needs. While adopting a single-dose vaccination schedule offers clear advantages in terms of affordability, cost-effectiveness, and expanded coverage, careful consideration is required for its integration into China’s existing healthcare infrastructure. At the same time, sustained investment in domestic vaccine development and centralized procurement strategies for imported vaccines will also be essential to alleviate the supply shortage. These insights are intended to deepen understanding of China’s vaccination strategy and support the transition toward a more efficient, accessible, and sustainable immunization framework. Ultimately, this review seeks to inform future health policy that enhances HPV vaccine uptake and accelerates China’s contribution to the global goal of eliminating cervical cancer by 2030.

## 2. Virology and Epidemiology of HPV

HPV belongs to the *Papillomaviridae* family, a group of viruses that have coexisted with the human species for millennia, undergoing minimal genetic changes over time. HPV particles measure approximately 60 nm in diameter and are composed of 72 pentameric capsomeres. Its genome consists of a single, circular, double-stranded DNA molecule of roughly 8000 base pairs [4]. The HPV genome encodes two key structural proteins: L1 and L2. L1, the major capsid protein, self-assembles into virus-like particles (VLPs) and induces type-specific antibodies, making it the foundation of HPV vaccine development. L2, a minor capsid protein, works in conjunction with L1 to form the virus capsid [5]. Based on L1 gene sequences, HPV classifications include distinct types, subtypes, and variants. A divergence of more than 10% in the L1 gene characterizes different HPV types, a 2–10% divergence indicates subtypes, and a divergence of less than 2% identifies variants [6]. HPV vaccines specifically target HPV infections at the type of level, with the International HPV Reference Center responsible for their classification. As of June 2023, 231 HPV types have been officially identified, with the latest discovery made by M. Rajeevan [7].

HPV types are categorized based on their tissue tropism and oncogenic potential. Cutaneous types, such as HPV 1, 4, and 5, are typically found in cutaneous and plantar warts, as well as some epithelial tumors. Meanwhile, mucosotropic types, including HPV 6, 11, 16, and 18, are frequently associated with lesions in the anogenital tract [8]. Based on their oncogenic risk, HPVs are classified as high-risk or low-risk. Low-risk variants, such as HPV 6 and 11, are predominantly associated with benign lesions, while high-risk types, including HPV 16, 18, 31, and 33, are significantly more likely to contribute to carcinogenesis [9]. While low-risk types are occasionally linked to neoplastic changes, high-risk types, particularly HPV 16 and 18, are frequently observed in cancers of the cervix (CIN), vulva (VIN), vagina (VaIN), penis (PIN), and anus (AIN) [10]. Most HPV infections are transient and resolve spontaneously within 1–5 years. However, persistent infection with high-risk types can lead to the development of precancerous lesions and invasive cancers.

HPV infection primarily targets epithelial cells and can result in a spectrum of pathological outcomes, ranging from benign warts to malignant cancers. While HPV infection is mostly discussed with cervical cancer, this group of viruses has broader oncogenic impact and is implicated in multiple other malignancies. Persistent HPV infection accounts for over 95% of cervical cancer cases, and high-risk HPV types also significantly contribute to vulval, vaginal, penile, anal, and head-and-neck cancers [11]. Globally, HPV 16 is the most prevalent oncogenic type, responsible for approximately 55% of cervical intraepithelial neoplasia (CIN) cases, followed by HPV 18, which accounts for about 18% of invasive cervical cancers (ICCs). Cumulatively, these two types cause over 70% of CIN cases worldwide. Other frequently detected types include HPV 33, 45, and 31, though in Asia, HPV 52 and 58 have higher prevalence rates [11].

HPV is one the most common sexually transmitted diseases, affecting individuals of all genders. While extensive research has focused on HPV-related diseases in women, particularly cervical cancer, male populations have received comparatively little attention due to the higher burden of HPV-related morbidity and mortality observed in women [12]. The highest rates of genital HPV infection are observed in young women under 25 years old, with a global prevalence of approximately 30%. A secondary peak in HPV prevalence commonly occurs in women aged 50–60 years, though regional variations exist [13]. Despite cervical cancer being a female-specific disease, HPV infection itself is not gender-restricted. Males are also susceptible to HPV-related disease burdens and serve as vectors in the transmission of the virus [12]. HPV transmission primarily occurs through skin-to-skin contact, including through microabrasions during sexual activity or through oral contact [14]. Risk factors for genital HPV acquisition include early initiation of sexual activity [9], multiple sexual partners, and exposure to partners with high-risk sexual behaviors. Young adolescents, immunocompromised individuals, and those with HIV are at a heightened risk of HPV infection.

Almost all cervical cancer cases are caused by persistent HPV infection, and the pathological progression from infection to malignancy has been well-established. Cervical cancer develops through four distinct phases: HPV transmission, viral persistence, the progression of infected cells into precancerous lesions, and eventual invasion into malignant cancer. These stages provide a clear framework for understanding disease pathogenesis and guiding prevention methods [1]. Interventions such as HPV vaccination and regular screening have proven crucial in reducing cervical cancer incidence. Primary screening methods, such as the HPV DNA test, colposcopy, Pap smear, biopsy, and acetic acid test, are effective tools for detecting abnormalities [14]. The World Health Organization (WHO) recommends initiating screening at age 30 (or age 25 for women living with HIV) to identify cervical abnormalities early, allowing for the timely treatment and prevention of cancer progression. While screening plays a vital role in detecting cervical cancer in its early stages, it does not prevent initial HPV infection and may be unaffordable in resource-limited settings. As such, HPV vaccination for both genders is the cornerstone of primary prevention, provides direct protection, and also enhances herd immunity and resilience to the disruption of cancer prevention [15]. The WHO emphasizes the safety, cost-effectiveness, and efficacy of HPV vaccines in reducing HPV-related infections, not only for cervical cancer cases but also other associated diseases.

## 3. Existing HPV Vaccines

HPV vaccines are developed based on virus-like particles (VLPs) derived from the L1 capsid protein of the virus [16]. These prophylactic vaccines are designed to prevent HPV infection, using non-infectious VLPs that contain no viral DNA, ensuring their safety for human use. Through the induction of humoral immunity, the vaccines generate type-specific antibodies to protect against infection. Globally, there are three main types of HPV vaccines based on their coverage: bivalent vaccines (HPV 16 and 18), quadrivalent vaccines (HPV 6, 11, 16, and 18), and nine-valent vaccines (HPV 6, 11, 16, 18, 31, 33, 45, 52, and 58) [17]. Currently, six licensed HPV vaccines are available worldwide, including three bivalent (Cervarix, Cecolin, and Walrinvax), two quadrivalent (Gardasil 4 and Cervaracx), and one nine-valent vaccine (Gardasil 9) [3]. Bivalent vaccines primarily target HPV 16 and 18, which are responsible for most cervical cancer cases, while nine-valent HPV vaccines offer broader protection against additional high-risk types. Among these options, Cervarix, Cecolin, and Gardasil 4 are widely available internationally, while Walrinvax is approved only in China, and Cervavac is exclusive to India as of 2022 [18].

HPV vaccines are highly effective, generating a strong and long-lasting immune response with robust immunological memory. Clinical trials have demonstrated that the serological response to HPV vaccination far exceeds that of natural infection. Vaccinated women remained seropositive for HPV 16 and 18 for up to 113 months, with consistently high levels of IgG antibodies. Notably, IgG levels in vaccinated individuals were 10.8-fold and 10.0-fold higher for HPV 16 and 18, respectively, compared to those generated through natural infection [19]. The vaccine offers long-term protection, with the efficacy of the three-dose regimen exceeding 90% for at least 10 years. Ongoing research has suggested continued effectiveness beyond this timeframe [20]. Studies in the United States further confirmed the vaccine’s impact, showing an 80.9% reduction in infections with four-valent vaccine-targeted HPV types and a 71% drop in infections with nine-valent HPV types among women who received at least one vaccine dose, compared to unvaccinated individuals [21].

HPV vaccines have consistently demonstrated a strongly safe profile, supported by both clinical trials and real-world post-marketing surveillance data [22]. Between 2006 and 2017, over 270 million doses were administered worldwide. Data from the Global Advisory Committee on Vaccine Safety indicated that serious adverse events were extremely rare, with anaphylaxis occurring at a rate of approximately 1.7 cases per million doses and occasional syncope as a frequent, stress-related response rather than a vaccine-specific effect. Other reported minor side effects—such as localized pain, swelling at the injection site, low-grade fever, or mild allergic reactions—were generally mild, self-limiting, and easily managed with proper care [23]. No evidence suggests any significant difference in the nature or severity of side effects between single-dose and multi-dose schedules.

## 4. Cervical Cancer and HPV Infection in the Globe and China

Cervical cancer is the fourth most prevalent cancer affecting women globally. In 2018, approximately 570,000 new cases and 311,000 deaths were reported. By 2022, these figures had increased to 662,301 new cases and 348,874 deaths, with projections suggesting a further rise to 700,000 cases and 400,000 deaths annually by 2030 [2]. The global burden of cervical cancer is expected to increase substantially without effective intervention. Notably, over 90% of cases occur in low- and middle-income countries, highlighting the health disparity [2]. Regionally, Asia bears the greatest burden, accounting for more than 58% of global cervical cancer cases and related deaths as of 2022, followed by Africa (20% of cases and 22% of deaths), Europe (10% of cases), and Latin America (10% of cases and 9% of deaths) [24]. According to the World Cancer Research Fund International, China and India together contributed approximately 40% of all new cervical cancer cases and deaths globally in 2022, highlighting their critical role in global cervical cancer prevention and control. China ranked first globally in cervical cancer incidence and second in mortality, underscoring gaps in prevention, early detection, and treatment initiatives (Figure 1) [25].

Recent data from the Global Cancer Burden Study in 2023 confirmed China as having the second-highest global cervical cancer incidence and mortality rates, with approximately 110,000 new cases and 59,000 deaths annually [26]. While many developed countries have experienced a significant decline in cervical cancer cases and deaths over the past three decades due to heightened awareness, widespread vaccinations, and effective screening programs, China continued to report rising rates. This increase could be attributed to insufficient screening coverage and the relatively late introduction of HPV vaccines and vaccination programs [27]. Between 2000 and 2019, studies reported that the overall prevalence of high-risk HPV infection among women in mainland China was 19.0% (95% CI: 17.1–20.9%), with the five most common high-risk subtypes identified as HPV 16, 52, 58, 53, and 18 [28]. HPV infection rates in China varied by geographic region. In East China, North China, Southeast China, and the Southwest, Northeast, and Northwest, the prevalence of HPV infection was 17.6% (95% CI: 15.6–19.7%), 23.8% (95% CI: 18.9–28.7%), 18.3% (95% CI: 13.4–23.3%), 17.5% (95% CI: 13.6–21.4%), 16.5% (95% CI: 11.2–21.8%), and 12.2% (95% CI: 10.0–16.7%), respectively [28].

HPV prevalence in China showed two distinct age peaks, among women aged ≤20–25 years and those aged 50–60 years, which is consistent with the global pattern of high-risk age groups. The WHO has emphasized the importance of prioritizing these vulnerable populations, a strategy that is particularly relevant for the Chinese context. This age pattern is consistent with the global HPV prevalence in high-risk age groups. Notably, HPV 16, 52, 58, 53, and 18 (ranking from high to low in prevalence) are the most commonly detected HPV genotypes in the Chinese population [29]. Among Chinese women diagnosed with cervical intraepithelial neoplasia (CIN), the overall HPV infection rate was reported at 84.37%. In cases of CIN1, the predominant HPV types detected were HPV 52 (20.31%), HPV 16 (16.81%), HPV 58 (14.44%), HPV 18 (6.44%), and HPV 53 (5.76%); in the CIN2/3 cases, HPV 16 was the most prevalent type (45.69%), followed by HPV 58 (15.50%), HPV 52 (11.74%), HPV 33 (9.35%), and HPV 31 (4.34%) [30]. Meta-analyses of global type-specific HPV prevalence have demonstrated that HPV 16, followed by HPV 18, is the most commonly associated with invasive cervical cancer [11,31]. However, research and reviews indicate that the top three prevalent high-risk HPVs in China are HPV18, 52, and 58. While HPV 16′s high prevalence aligns with the global pattern, the prominence of HPV 52 and 58 highlights significant regional differences in the distribution of high-risk HPV types in China, compared to the global infection pattern.

## 5. HPV Vaccination

The World Health Organization (WHO) has established a global strategy to accelerate the elimination of cervical cancer through collaborative efforts. The aim is to reduce annual cervical cancer cases to four or fewer per 100,000 women and to enhance vaccination, screening, and treatment by 2030. Among these interventions, vaccination is considered the most effective method for eliminating cervical cancer, as it provides widespread, long-term protection and is cost-efficient, especially in large-scale immunization campaigns [2]. The effectiveness of the vaccine is the highest when administered before HPV exposure. Vaccinating individuals before their first sexual contact can prevent over 90% of targeted HPV-related infections, abnormalities, and precancerous lesions, whereas vaccination after HPV exposure offers protection against only 50–60% of infections [32]. As such, the WHO recommends prioritizing routine HPV vaccination for girls aged 9–14 years, ideally before sexual activity begins (Table 1).

HPV vaccines are administered following a standardized procedure to ensure both safety and immunogenic efficacy. Each dose consists of 0.5 mL, delivered via intramuscular injection, preferably in the deltoid muscle of the upper arm or alternatively in the anterolateral thigh [33]. Pre-vaccination screening, including Pap smears and HPV DNA testing, is not required before vaccine administration [34]. For multi-dose schedules, HPV vaccine administration should adhere to the recommended dosing intervals: the two-dose schedule requires a minimum interval of five months between doses, while the three-dose schedule involves intervals of at least four weeks between the first and second doses, twelve weeks between the second and third doses, and a minimum of five months between the first and final dose [35]. Recognizing the burden of HPV-related diseases and the need for clear vaccination guidelines in large-scale immunization programs, the WHO continually updates the recommendations for the HPV vaccination dose schedule. The HPV vaccination program was first introduced in 2009, which was an initial three-dose schedule for young adolescent girls, emphasizing the use of a consistent vaccine type across all doses [36]. In 2014, the WHO revised its recommendation based on epidemiological evidence, highlighting the feasibility of a reduced-dose schedule. Guidelines in 2014 specified a two-dose schedule for girls aged 9–14, while retaining the three-dose regimen for individuals aged 15 and older or for cases where the second dose was administered earlier than 6 months after the first dose [35]. In December 2022, the WHO released the latest update, recommending a single-dose HPV vaccination schedule due to evidence showing its comparable efficacy and durability of protection compared to multi-dose regimens. This change aims to improve vaccination accessibility, lower costs, and expand coverage. The current WHO guidelines now recommend a one- or two-dose schedule for girls aged 9–20, two doses with a 6-month interval for women over 21, and at least two doses (or three doses when possible) for immunocompromised individuals, including those living with HIV [3] (Table 1). From 2009 to 2022, the recommended HPV vaccine schedule has continued evolving, reflecting growing evidence from clinical research, real-world data on vaccine effectiveness, and the need to improve coverage and logistical feasibility.

## 6. HPV Vaccination in China

Vaccination plays a crucial role in reducing the prevalence of HPV infections and cervical cancer. While it has been in clinical use for over 15 years globally, its widespread application in mainland China has occurred only recently. Five prophylactic HPV vaccines have been approved by the China Food and Drug Administration (CFDA), including the domestically produced bivalent vaccines Cecolin and Walrinvax, the imported bivalent vaccine Cervarix, the imported quadrivalent vaccine Gardasil 4, and the imported nine-valent vaccine Gardasil 9 [37]. Cervarix, the first bivalent HPV vaccine to enter the Chinese market, was approved in 2016 as a self-funded option for individuals aged 9 to 45 [38]. Gardasil 4 was approved by the CFDA in 2017 for individuals aged 20 to 45, and Gardasil 9 was approved in 2018 for individuals aged 11 to 26 [28]. Two domestically manufactured vaccines, Cecolin and Walrinvax, were later approved, with Cecolin receiving CFDA approval in 2019 and WHO qualification in October 2021. Walrinvax, approved in 2022, is currently undergoing WHO prequalification review [39,40].

In China, receiving an HPV vaccine is voluntary and self-funded. The standardized vaccine price per dose in China is listed as follows (unpublished observation, manuscript in preparation):Cecolin (bivalent, domestic): RMB 349 per dose;Cervarix (bivalent, imported): RMB 598 per dose;Gardasil 4 (quadrivalent, imported): RMB 798 per dose;Gardasil 9 (nine-valent, imported): RMB 1298 per dose;Service fee for vaccine administration: RMB 20 per dose.

Globally, HPV vaccines are priced at USD 4.5–4.6 per dose in Gavi-supported countries (for quadrivalent and bivalent vaccines, respectively). However, in China, the cost is significantly high, ranging between USD 43 and USD 187 per dose for individuals and about USD 35 per dose for government procurement. The nine-valent HPV vaccine is particularly expensive, costing up to USD 187 per dose, making it inaccessible for the majority of eligible women and unaffordable for government-funded programs [41,42]. Although some pilot programs and local government initiatives provide free vaccinations, these programs are localized and offer only a limited range of vaccine types. Many individuals pay out of pocket for the nine-valent vaccine, which has grown in popularity following the extension of its age eligibility to 9–45 years. However, demand for this vaccine continues to exceed supply [37]. As of 2024, HPV vaccines have not yet been included in China’s National Immunization Program [43].

In response to the WHO’s cervical cancer elimination initiative, China has introduced policies such as the Healthy China Action (2019–2030) to increase HPV vaccine accessibility. In November 2020, the CFDA lowered the minimum age for quadrivalent HPV vaccination to 9 years [44]. In 2021, the Health City Innovation Pilot Program was launched to support comprehensive cervical cancer prevention and control, including the introduction of government-funded HPV vaccination programs in certain regions with adequate resources [45]. In January 2023, China reaffirmed its commitment to expanding HPV vaccine coverage through the Action Plan for Accelerating the Elimination of Cervical Cancer (2023–2030). This plan encourages school-age HPV vaccination programs with financial subsidies to improve affordability and vaccination rates among adolescent girls. In July 2024, the National Health Commission indicated that the decision to include HPV vaccines in the national immunization program would depend on the results of rigorous assessments regarding the vaccine’s safety, efficacy, accessibility, and necessity [46].

In line with the WHO’s revised HPV vaccination guidelines issued in 2022, China has started updating its own vaccination schedules. For example, Jiangsu Province launched the Healthy China Initiative Innovation Model Pilot Program, which provides free HPV vaccinations in certain areas. In Nanjing, Wuxi, and Lianyungang, free two-dose HPV vaccinations with the domestic bivalent vaccine, Cecolin, were offered to middle school girls [47]. Suzhou further expanded this program in 2023 by providing fully government-funded, two-dose Cecolin vaccinations to all seventh-grade girls. A partial subsidy was also introduced for the nine-valent HPV vaccine, reducing costs by RMB 200 per dose, with the total cost for the three-dose regimen amounting to RMB 600 [48]. In March 2024, Jiangsu Province updated its immunization guideline to allow girls aged 9–14 to receive a two-dose schedule (0 and 6 months, with a minimum 5-month interval) as an alternative to the standard three-dose regimen [49]. Nationally, the standard practice in China remains the three-dose schedule for females aged 9–45, regardless of vaccine type. Doses are administered at 0, 2, and 6 months, with a 1- to 2-month interval between the first and second doses and a 4-month interval between the second and third doses. For girls aged 9–14 using the domestic bivalent Cecolin or the imported nine-valent Gardasil 9, a two-dose schedule is an option if the two doses are administered six months apart. For those who have already received two doses less than five months apart, a third dose is required (Table 2).

Between 2017 and 2022, a total of 85.79 million HPV vaccine doses were administered across China, with annual uptake increasing significantly—from 44,000 doses in 2017 to over 47.2 million doses in 2022 [42,50]. Despite this progress, vaccine coverage remains low. In 2020, only 3% of women aged 9–45 in China had completed the full HPV vaccination series [51]. By the end of 2022, cumulative first-dose coverage had increased to 10.15%, and third-dose coverage had increased to 6.21%, both of which remain below global averages and the threshold for herd immunity [50]. According to the National Bureau of Statistics, 80 million vaccine doses are required annually to achieve 90% coverage among the targeted girl aged 9–14, yet in 2022, manufacturers supplied only about 59 million doses (30 million bivalent, 14 million quadrivalent, and 15 million nine-valent doses) [42]. The shortfall of over 15 million doses per year, combined with high vaccine costs, continues to hinder the accessibility and uptake of HPV vaccines in China. Therefore, addressing these challenges is essential for meeting the growing demand for HPV vaccines and achieving cervical cancer prevention goals.

## 7. Transitioning from a Three-Dose to a Single-Dose HPV Vaccination Program: Challenges and Opportunities

Support for a three-dose HPV vaccination schedule in China stems from its superior immunogenicity and longer protective duration compared to single-dose regimens. Clinical trials and immunological studies have shown that the three-dose schedule generates higher antibody titers, resulting in more robust and sustained immunity that covers a broader range of HPV strains. A study by the U.S. CDC, as part of the HPV-IMPAC project, which analyzed data from 3300 women aged 18–39 years (2008–2014) vaccinated with different HPV dose regimens, revealed that the adjusted odds ratios (ORs) for HPV 16/18-positive CIN2+ cases were 0.53 (95% CI: 0.37–0.76) for one dose, 0.45 (95% CI: 0.30–0.69) for two doses, and 0.26 (95% CI: 0.20–0.35) for three doses administered at least 24 months before screening [52]. While all schedules offer significant protection against HPV infection compared to no vaccination, the three-dose regimen demonstrated nearly 40% greater effectiveness, reinforcing its superior protective benefits (Figure 2).

The willingness to opt for specific types of HPV vaccines and pay for them varies based on socio-demographic factors. Studies have found that individuals with higher socioeconomic status, higher education levels, younger age, and living in urban areas show greater willingness to pay and are more likely to receive HPV vaccination [53]. Female college students in Zhejiang Province, for example, demonstrated a strong willingness to pay for the full regimen of the nine-valent HPV vaccine, valuing its higher level of protection. In one study, the willingness rates for bivalent vaccines (Cecolin and Cervarix), quadrivalent vaccine Gardasil 4, and nine-valent vaccine Gardasil 9 were 11%, 10%, 22%, and 58%, respectively [54]. Increasingly, people in China are seeking the more protective quadrivalent and nine-valent vaccines, despite their higher costs and limited availability [55]. With the approved age for nine-valent vaccines expanding to 9–45 years in 2022, the demand for Gardasil 9 has surged, far outstripping supply [37].

Emerging scientific evidence supports the potential of a single-dose HPV vaccination schedule. Research from immunogenicity trials, efficacy studies, and real-world data shows that a single dose provides strong protection against HPV infection, comparable to multi-dose regimens. For instance, at 18 months post-vaccination, the efficacy of a single dose against persistent high-risk HPV16/18 infection was 97.5% (95% CI: 82–100) for both nine-valent and bivalent vaccines [56]. While the immune response from single-dose recipients was lower than that of two- or three-dose recipients, antibody levels remained stable over four years. Over seven years of follow-up, the occurrence of cumulative and persistent HPV16/18 infections in single-dose recipients was similarly low compared to multi-dose recipients, while unvaccinated individuals showed significantly higher infection rates [57]. Studies indicate that two doses of the bivalent vaccine in adolescent girls elicit immune responses comparable to three doses in adults [58]. High vaccine effectiveness regardless of the number of doses is also confirmed when the first dose was administered to women aged 18 or younger [59]. Long-term studies, such as the DoRIS study in Tanzania, have observed consistent protection from a single dose for up to 11 years [60].

The WHO’s 2022 HPV vaccine market study anticipated a significant increase in global HPV vaccine demand over the coming decade (2022–2031), driven by the expansion of vaccination programs following the Call to Action for cervical cancer elimination [61]. The report highlighted the impact of dose schedules on global supply–demand dynamics. For instance, a shift to single-dose schedules could substantially reduce total vaccine requirements, potentially stabilizing demand at approximately 70 million doses annually by 2028. On the other hand, implementing gender-neutral two-dose regimens in all high-income and middle-income countries, especially if boys were included, could increase global annual demand to over 150 million doses by 2030 [62]. To meet the global cervical cancer elimination target, the WHO estimated that over 160 million HPV vaccine doses would be required annually to achieve at least 90% coverage across all countries [62]. Given that the single-dose schedule has demonstrated comparable efficacy and long-term protection, its adoption offers critical flexibility in global vaccination strategies. A simplified schedule could help reserve doses for vulnerable populations and support the implementation of catch-up programs when resources allow. This approach would enable more efficient allocation of limited vaccine supplies, ultimately optimizing population-level coverage and strengthening HPV-related disease prevention efforts.

Adopting a single-dose HPV vaccination schedule in China could be a cost-effective and operationally feasible way to optimize vaccine allocation and improve vaccine coverage (Figure 2). Modeling studies indicate that a single-dose regimen would be the most cost-efficient option, while the three-dose nine-valent HPV vaccine regimen would maximize benefits if vaccine costs were halved [63]. Even under conservative assumptions of 30 years of protection from a single dose, most cervical cancer cases could still be averted, making the single-dose strategy both practical and cost-saving [39]. Catch-up vaccination for older cohorts under this schedule has also been shown to enhance cost-effectiveness. For instance, reallocating the second dose to vaccinate 20-year-olds would lead to net savings of 4.1–6 billion while maximizing the program’s efficiency [64]. Operationally, a single-dose schedule simplifies vaccination logistics and reduces the burden on healthcare delivery systems, thereby improving accessibility and accelerating coverage expansion [65]. HPV vaccine coverage in China remains relatively low. As of 2022, the first-dose coverage rate was only 10.15%, significantly below global averages and far from sufficient to achieve the goal of cervical cancer elimination [50]. Shifting to a single-dose schedule could mitigate supply constraints, enabling more rapid achievement of high coverage rates essential for population-level protection. At the same time, the flexibility afforded by single-dose scheduling allows greater adaptability in national immunization strategies, such as supporting gender-neutral vaccination or reserving multi-dose regimens for immunocompromised populations.

From an epidemiological perspective, promoting single-dose vaccination with the nine-valent HPV vaccine may offer greater protective benefits in China due to its distinct HPV genotype distribution. Epidemiological data from China reveal a unique HPV genotype distribution, with HPV 52, 58, and 18 showing high prevalence rates [11]. While bivalent and quadrivalent vaccines primarily target HPV 16 and 18, they may inadequately address the broader range of high-risk genotypes circulating in China. This mismatch potentially limits their population-level protective impact. As single-dose regimens have demonstrated comparable efficacy and duration of protection to multi-dose regimens, prioritizing the nine-valent vaccine in a single-dose schedule could enhance the coverage of locally prevalent high-risk HPV types. This approach would broaden the spectrum of protection, offering a more epidemiologically relevant and equitable solution for cervical cancer prevention in China.

Vaccination is the most effective strategy for preventing HPV infections and related diseases. However, HPV vaccine coverage in China remains relatively low. As of 2022, the first-dose coverage rate was only 10.15%, significantly below global averages and far from sufficient to achieve the goal of cervical cancer elimination. In response to the WHO’s 2022 position paper, China has now entered a critical transition phase, with ongoing discussions and policy adjustments aimed at adopting a reduced-dose vaccination schedule. The current national immunization schedule in China continues to align more closely with the WHO’s 2014 recommendations than with the updated 2022 guidelines. Under the existing policy, girls aged 9–14 are eligible for an optional two-dose regimen, while a three-dose schedule remains the standard for females aged 15–45, regardless of vaccine type. In contrast, the 2022 WHO recommendation supports a more simplified approach: one or two doses for individuals aged 9–20, and two doses for women older than 21. In China, the optional two-dose schedule is currently limited to the domestic bivalent vaccine (Cecolin) and the imported nine-valent vaccine (Gardasil 9) and only applies to girls aged 9–14. This restricted application underscores a significant policy gap between China’s current practice and the WHO’s latest guidance.

The persistently high cost of HPV vaccines remains a significant barrier to expanding vaccine coverage in China. As previously mentioned, the price of the HPV vaccine in China is higher than the global average. Despite the financial support, subsidies are typically restricted to those receiving vaccines through school-based immunization campaigns. Outside these programs, individuals must bear the full cost out of pocket. The nine-valent HPV vaccine Gardasil 9 is the most preferred yet the most expensive option, whose price might remain unaffordable for certain populations even with government subsidies. This financial barrier may limit vaccine uptake, especially among lower-income families, potentially exacerbating disparities in HPV vaccine coverage.

Vaccine supply shortages, particularly regarding the nine-valent HPV vaccine, pose another major challenge. Currently, the available HPV vaccines in China fall short of the estimated requirements, with over 15 million additional doses needed to cover 90% of girls aged 9–14 years [42]. Annual vaccine production and distribution remain inadequate to meet national demand, a problem intensified by the recent expansion of eligibility criteria. Gardasil 9, introduced to mainland China in 2018 for individuals aged 11 to 26 years, whose approved age range was expanded to 9 to 45 years in 2022 [28]. This policy change significantly increased the eligible population, amplifying demand for this broad-spectrum vaccine and further straining the limited supply [37]. Moreover, the lack of routine male HPV vaccination further compounds the vaccine supply shortage. Adopting a gender-neutral HPV vaccination strategy could reduce population-level transmission, combat misinformation, minimize vaccine-related stigma, and promote gender equity [66]. As of 2019, gender-neutral HPV vaccination was recommended in only 32 countries, while most others limited their efforts to vaccinating females. [67]. Similarly, China’s vaccination efforts remain largely focused on girls. However, gender-neutral HPV vaccination plays a critical role in controlling transmission and contributes to herd immunity, as an effective strategy to improve the resilience to the disruption of cancer prevention and overall disease burden reduction [15]. If vaccination targets in China are further expanded to include males, vaccine demand will increase substantially. Additionally, the evolving vaccination guidelines, though reflective efforts to align with WHO recommendations, can create confusion among healthcare providers as well as the public. Frequent updates to dose schedules and eligibility criteria, particularly during this transitional phase, may hinder consistent implementation and compromise the clarity and effectiveness of vaccination efforts.

Adopting a single-dose strategy could help overcome many existing barriers while increasing HPV vaccination coverage across China. A single-dose schedule would significantly reduce out-of-pocket costs by lowering both the number of required doses and the associated vaccine and administration fees. This approach enhances affordability, alleviates vaccine supply constraints, and allows for more flexible resource allocation. Additional strategies, including increased investment in domestic vaccine research and development, centralized procurement of imported vaccines, and integration of HPV vaccination into the National Immunization Program (NIP), are also critical for policy development. Effective vaccine procurement strategies are essential to ensure sustainable vaccine supply and price [68]. Centralized procurement could enhance the government’s purchasing power, facilitating price negotiations with manufacturers and making vaccines more affordable, particularly for countries without Gavi support [69]. To address supply shortages, we recommend prioritizing domestic vaccine production alongside centralized procurement of imported vaccines to increase national vaccine availability. Collectively, these strategies offer significant opportunities to strengthen HPV vaccination efforts and improve public health outcomes in China.

China is at a transformative juncture in its HPV vaccination efforts. While challenges such as high vaccine costs, limited supply, and policy inconsistencies remain, the country is making progress toward aligning with the WHO’s cervical cancer elimination goals. Transitioning to simplified vaccination strategies, such as a single-dose schedule, offers an opportunity to lower barriers, increase coverage, and protect more women and girls from HPV-related diseases. With targeted investments and policy reforms, China can achieve significant strides in cervical cancer prevention and improve public health outcomes.

## 8. Conclusions

By examining the virology, epidemiology, existing vaccines, and vaccination strategies, this paper underscores that cervical cancer, while largely preventable through HPV vaccination, remains a significant global health challenge. This burden is unequally distributed, disproportionately affecting women in low- and middle-income countries such as China, India, and many nations across sub-Saharan Africa. Beyond health concerns, this inequity raises issues of health justice. In response to the rising incidence and mortality of cervical cancer, the WHO has taken a leading role in shaping global vaccination strategies, regularly updating recommendations, and recently endorsing a reduced-dose schedule to accelerate progress toward cervical cancer elimination. These global efforts align closely with the principles of the 2030 Agenda for Sustainable Development, which emphasizes “leaving no one behind,” promoting health equity, and advancing gender equality as fundamental human rights.

In summary, this study provides timely and valuable insights into HPV vaccination practices in China, with a focus on key policy developments. The findings indicate that China is undergoing a significant transition in its HPV vaccination strategy, progressively moving toward reduced-dose regimens while steadily expanding vaccine coverage. However, current policies remain more closely aligned with the WHO’s 2014 recommendations rather than the updated 2022 guidelines, which advocate for a single-dose schedule. The current adoption of a two-dose regimen for certain age groups serves as an interim step, but further adjustments will likely be necessary to fully align with global recommendations and optimize vaccine access. While there has been notable progress in expanding vaccination coverage, challenges remain in addressing HPV vaccine shortfall and high vaccine prices. Transitioning to a single-dose schedule, supported by strong scientific evidence and economic modeling, represents a promising opportunity to improve affordability, optimize vaccine coverage, and accelerate progress toward cervical cancer elimination. Overcoming the existing challenges will also require sustained policy innovation, centralized procurement of imported vaccines, expansion of public funding mechanisms, and increased investment in domestic vaccine development and manufacturing capacity. China’s experience offers valuable lessons for other countries navigating similar transitions in HPV vaccination policy.

This study, however, has several limitations. First, the absence of real-world implementation data, such as direct interviews with policymakers or more detailed vaccine distribution statistics, limits the ability to fully evaluate the feasibility of the proposed policy changes. Second, this study primarily focuses on policy in Jiangsu Province, which, while allowing for a deeper contextual understanding, constrains the applicability of findings to other regions of China. The lack of comparative data from other provinces, particularly rural and underserved areas, risks oversimplifying the significant geographical disparities in policy adoption and implementation across the country. Third, this review focuses primarily on cervical cancer prevention in females, overlooking the broader public health impact of HPV infection. While HPV is a significant health issue affecting all genders, current prevention efforts remain largely centered on cervical cancer, with limited attention given to its role in other HPV-related cancers and diseases [70]. The role of male vaccination is also underexplored, even though including boys in HPV vaccination strategies could enhance herd immunity, reduce virus transmission, and protect against HPV-related diseases in all genders [71].

Future research should aim to address these gaps by expanding the geographic and demographic scope of analysis and incorporating qualitative methods, such as interviews with national and local health officials, policymakers, and frontline healthcare professionals. These firsthand insights could offer a richer understanding of implementation challenges, logistical hurdles, and public perceptions, thereby informing evidence-based policymaking. Further evaluation of school-based delivery, public education strategies, and gender-neutral vaccination could support broader and more equitable vaccine access. Research exploring the prevention of non-cervical HPV-related diseases and the benefits of male vaccination will be critical for informing a more comprehensive, inclusive HPV control strategy. Together, these efforts can strengthen China’s contribution to global cervical cancer elimination and HPV-related disease prevention.

## Figures and Tables

**Figure 1 vaccines-13-00786-f001:**
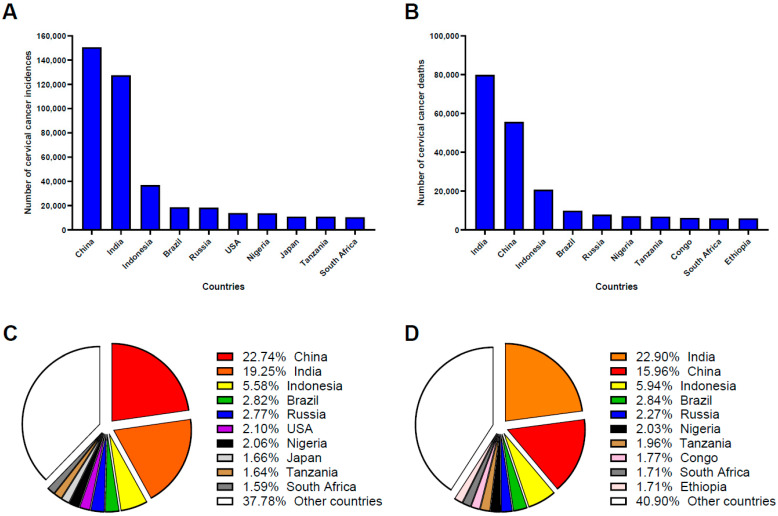
Cervical cancer incidence and mortality statistics for 2022. Data were compiled from surveillance reports by the World Health Organization (WHO) and the World Cancer Research Fund (accessed on 2 June 2025). Data represent the female population, as cervical cancer is a sex-specific malignancy that occurs exclusively in individuals with a cervix. (**A**) The top 10 countries with the highest cervical cancer incidence rates, along with their corresponding case numbers. (**B**) The top 10 countries with the highest cervical cancer mortality rates, along with their associated death counts. (**C**) Global distribution of cervical cancer incidence rates by country. (**D**) Global distribution of cervical cancer mortality rates by country. Data presented in (**C**,**D**) are expressed as the percentage of cases by country, calculated as percentage of cases = 100 × (number of cases/total number of cases reported globally). Other countries include all reported countries (175 countries) except the top 10. The pie charts were generated using GraphPad Prism 10.3.1 software.

**Figure 2 vaccines-13-00786-f002:**
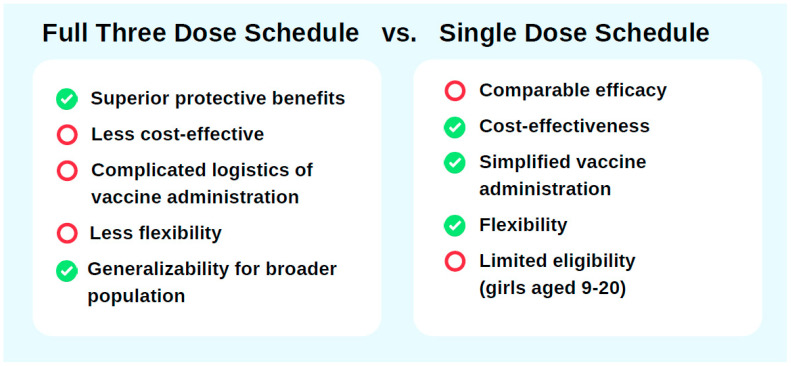
Comparison between full three-dose and single-dose HPV vaccination schedules across key implementation dimensions. This figure highlights a comparison of the full three-dose HPV vaccination schedule with the single-dose regimen, framed within current scientific and policy discussions about the global shift toward simplified vaccination strategies. It examines five critical dimensions: protective efficacy, cost-effectiveness, vaccine administration, scheduling flexibility, and generalizability. While the three-dose schedule has long been the standard in HPV immunization programs, emerging evidence, supported by the World Health Organization’s (WHO) 2022 HPV vaccination update, advocates for a single-dose schedule in specific age groups. The single-dose regimen demonstrates comparable immunogenicity in target populations, improved cost efficiency, and greater feasibility, especially in resource-limited settings. This comparison underscores the trade-offs between the two strategies and aims to inform future policy decisions on HPV vaccine delivery, particularly in countries undergoing a transition in their immunization strategies, such as China. The green tick icon represents the advantages from a particular approach, while the red circle icon indicates its drawbacks.

**Table 1 vaccines-13-00786-t001:** Summary of WHO position paper updates on HPV vaccination schedules (2009, 2014, and 2022), including recommended dose regimens, vaccine types, and target age groups.

WHO Updates	HPV Vaccination Schedule		
2009	2v	young adolescent girls	3-dose schedule (0, 1, and 6 months)
	4v	young adolescent girls	3-dose schedule (0, 2, and 6 months)
2014	2v	girls aged 9–14	2-dose schedule (0 and 6 months)
	The second dose can be given between 5 and 7 months after the first dose		
	4v	girls and boys aged 9–13	2-dose schedule (0 and 6 months)
			3-dose schedule (0, 2, and 6 months)
	If the second vaccine dose is administered earlier than 6 months after the first dose, a third dose should be administered.		
	2v/4v	over 15 years old	3-dose schedule (0, 2, 6 months)
2022	2v/4v/9v	girls aged 9–20	1 or 2-dose schedule (0, 6 months)
		women older than 21	2-dose schedule (0, 6 months)
		immunocompromised and/or HIV-infected	a minimum of 2 doses, and when feasible, 3 doses remain necessary

During the vaccination process, the same vaccine type is administered for the multi-dose regime. “2v,” “4v,” and “9v” refer to the bivalent, quadrivalent, and nine-valent HPV vaccines, respectively. The timeline indicated (for instance, 0, 2, and 6 months) corresponds to the standard multi-dose schedule: the second dose is typically administered one to two months after the first dose, and the third dose approximately six months after the initial dose—equating to about four months after the second dose.

**Table 2 vaccines-13-00786-t002:** Summary of China’s current HPV vaccination schedule, including recommended dose regimens, vaccine types, and target age groups.

Current Schedule In China	2v/9v	Girls aged 9–14	An optional 2-dose schedule (0 and 6 months)
		* For the schedule transition, girls aged 9–14 who have already received two doses of the HPV vaccine, if the time interval between the first dose and second dose is >5 months, a third dose is not required; otherwise, a third dose is required	
		Female aged 9–45	3-dose schedule (0, 2, and 6 months)
	4v	Female aged 9–45	3-dose schedule (0, 2, and 6 months)

* During the vaccination process, the same vaccine type is administered for the multi-dose regime. “2v,” “4v,” and “9v” refer to the bivalent, quadrivalent, and nine-valent HPV vaccines, respectively. The timeline indicated (for instance, 0, 2, and 6 months) corresponds to the standard multi-dose schedule: the second dose is typically administered one to two months after the first dose, and the third dose approximately six months after the initial dose—equating to about four months after the second dose.

## Data Availability

No new data were created in this review article.

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
