# Peer review of "Literature Review and Policy Recommendations for Single-Dose HPV Vaccination Schedule in China: Opportunities and Challenges"

_vaccines, 2025, doi:10.3390/vaccines13080786_

Round 1
Reviewer 1 Report
Comments and Suggestions for Authors
1. Inappropriate period punctuation appears multiple times in the text. Check and revise it.
2. Abstract need more recent information regarding this aim of review.
3. What is the novelty of your research ? No clear direction.
4. you donot need to repeat several time full form for example HPV line 63. Please check others also.
5. It would be interesting if you add sex and age in the figure 1.
6. why your data only focus on WHO ? why not others UNICEF, different NGO data.
7. You should tell us the side effect of single or full dose ?
8. You should more information regarding Vaccine supply and modelled scenarios.
9. In the figure 1 please add more data regarding target age vaccination vs non vaccination.
10. Even through it is more important to know one dose, two dose, full dose differenc among the group of people.
11. Why you did not add information related economic purpose ? what is the total cost ? make a bar graph.
12. Do you have any data regarding Long term efficacy and safety of HPV vaccines ?
13. Title is not clear please modify it.
14. The introduction should better explain the rationale and clearly state the knowledge gap this study aims to address.
15. There are inappropriate uses of periods throughout the manuscript. Please review and revise punctuation.
16.Verb tenses should be consistent—ideally, use the past tense throughout.
17. The manuscript’s grammar requires improvement. Have you considered professional proofreading or English-language editing services?
Author Response
Thank you very much for taking the time to review this manuscript. We have responded to all the pointers and have revised the manuscript (text in black is your original comments, text in blue is the response and revised contents). We have also updated the reference list. Please find the detailed responses below and the corresponding revisions/corrections highlighted in the re-submitted manuscript.
- Inappropriate period punctuation appears multiple times in the text. Check and revise it.
Response: Thank you for your suggestions! We have double-checked the period punctuation and revised them.
- Abstract need more recent information regarding this aim of review.
Response: Thank you for your advice! We have revised the abstract and added the objective of this study in this section, as well as a more detailed description in the introduction section.
“The objective of this review is to critically examine the current HPV vaccination landscape in China, including vaccination policies, immunization schedules, supply–demand dynamics, and the feasibility of transitioning to a single-dose regimen. By synthesizing recent developments in HPV virology, epidemiology, vaccine types, and immunization strategies, we identify both opportunities and barriers unique to the Chinese context” (Line 13-18)
- What is the novelty of your research? No clear direction.
Response: Thank you for your instructive comment! We have added the novelty to the introduction section, together with the revision regarding the knowledge gap and rationale.
“This review uniquely addresses that gap by providing a comparative analysis of HPV epidemiology and vaccination strategies between the Globe and China’s latest situation. Special emphasis is placed on the implications of the WHO’s 2022 recommendation supporting single-dose vaccination, assessing its applicability and potential benefits within the China’s context.” (Line 65-69)
- you do not need to repeat several time full forms for example HPV line 63. Please check others also.
Response: Thank you for your careful review! We avoided unnecessary repetition of full forms once acronyms are defined. In the revised version, we standardized the use of “Human Papillomavirus” by spelling it out only at its first mention (Line 40) and subsequently using the abbreviation “HPV.” When referring to vaccine types, we use the format “HPV [variant number]” (e.g., HPV 16, HPV 18). When listing several HPV variants at the same time, we mention “HPV” only once, followed by the additional type numbers (e.g., HPV 18, 52, and 58) to avoid unnecessary repetition. We standardized vaccine names, using “Gardasil 9” to refer to the nine-valent vaccine and “Gardasil 4” for one of the two quadrivalent vaccines, in line with the official product names used by their manufacturer—Merck Sharp & Dohme.
- It would be interesting if you add sex and age in the figure 1.
Response: Thank you for your thoughtful suggestion. Cervical cancer is a female-specific disease, as it affects the organ—cervix, that is a part of the female reproductive system. Males do not have a cervix, and they won’t have cervical cancer. To avoid potential confusion, we have mentioned the gender in the figure legend. Also, we have revisited the data source; we did not find any exposed information on age of incidence and mortality.
“Figure 1. Cervical cancer incidence and mortality statistics for 2022. Data were compiled from surveillance reports by the World Health Organization (WHO) and the World Cancer Research Fund (accessed on 2 June 2025). All data represent the female population, as cervical cancer is a sex-specific malignancy that occurs exclusively in individuals with a cervix.” (Line 212-213)
Regarding age-specific data, we revisited the data sources; however, no information on age-related incidence or mortality was publicly available. This issue would be further addressed in the response to comment 6.
- why your data only focus on WHO? why not others UNICEF, different NGO data.
Response: Thank you for this thoughtful comment. We fully agree that incorporating data from multiple sources such as UNICEF or other NGOs would enrich the analysis. The purpose of figure 1 is provide a broad overview of cervical cancer incidence and mortality, demonstrating the unevenly distributed cervical cancer burden, particularly in developing countries and China, which is the primary focus of this study. During the manuscript preparation, we actively sought out global datasets that could provide recent and comprehensive data regarding cervical burden.
While some sources, including national and sub-national databases, offer detailed information such as age at diagnosis or vaccination coverage, these data are often limited to specific provinces or countries. In contrast, the World Health Organization (WHO) provides the most comprehensive and standardized global statistics currently available. For this reason, we relied on these sources to ensure data comparability and accuracy across countries. We also cross-verified WHO statistics with data from the World Cancer Research Fund (WCRF) to enhance reliability.
We acknowledge the limitations of using a single primary source and hope that more integrated and updated global datasets will become available in the future, including from UNICEF and other international partners.
- You should tell us the side effect of single or full dose?
Response: Thank you for your comment! The safety profile of HPV vaccines is excellent, regardless of whether a single-dose or multi-dose regimen is administered. The rate of severe adverse events, such as anaphylaxis, is extremely low. As noted in the manuscript, “Other reported minor side effects—such as localized pain, swelling at the injection site, low-grade fever, or mild allergic reactions—were generally mild, self-limiting, and easily managed with proper care.” (Line 189-191)
“No evidence suggests any significant difference in the nature or severity of side effects between single-dose and multi-dose schedules.” (Line 191-192) We have added an illustration addressing side effects by dose schedule in Section 3, Existing HPV Vaccines.
- You should more information regarding Vaccine supply and modelled scenarios.
Response: Thank you very much for your valuable feedback! We acknowledge the importance of providing more detailed information on vaccine supply and modelled scenarios, particularly in illustrating how the adoption of a single-dose schedule could optimize vaccine allocation. In response, we have revised and expanded the discussion on vaccine demand–supply dynamics in Section 7 to address these aspects more comprehensively.
“For instance, a shift to single-dose schedules could substantially reduce total vaccine requirements, potentially stabilizing demand at approximately 70 million doses annually by 2028. On the other hand, implementing general-neutral two-dose regimens in all high-income and middle-income countries, especially if boys were included, could increase global annual demand to over 150 million doses by 2030 (62). To meet the global cervical cancer elimination target, the WHO estimated that over 160 million HPV vaccine doses would be required annually to achieve at least 90% coverage across all countries (62). Given that the single-dose schedule has demonstrated comparable efficacy and long-term protection, its adoption offers critical flexibility in global vaccination strategies. A simplified schedule could help reserve doses for vulnerable populations and support the implementation of catch-up programs when resources allow. This approach would enable more efficient allocation of limited vaccine supplies, ultimately optimize population-level coverage and strengthen HPV-related disease prevention efforts.” (Line 459-471)
- In the figure 1 please add more data regarding target age vaccination vs non vaccination.
Response: Thank you for your valuable suggestion. We agree that incorporating data on cervical cancer incidence and mortality among vaccinated versus non-vaccinated individuals across different age groups would enhance the comprehensiveness of Figure 1. However, after a thorough review of available surveillance sources, we were unable to obtain stratified data that cooperating global cervical cancer outcomes by vaccination status and target age groups at a global scale. We acknowledge the importance of this information and have noted this as a limitation in the manuscript. We hope that as more countries integrate HPV vaccination into their national immunization programs and longitudinal data become available, such stratified analyses will be possible in future studies.
- Even through it is more important to know one dose, two dose, full dose difference among the group of people.
Response: Thank you for your comment! The differences in dose schedule, including dose number, time interval between doses, vaccine type and suitable population have been presented in table 1.
From 2009 to 2022, the recommended HPV vaccine schedule kept evolving, changing from the original three-dose regimen to the current option of a single-dose schedule. This reflects growing evidence from clinical research, real-world data on vaccine effectiveness, and the need to improve coverage and logistical feasibility. This flexibility allows countries to tailor their vaccination strategies according to local health system capacity, population needs, and available resources. Although the three-dose schedule may offer slightly enhanced immunogenicity, evidence indicates that the single-dose regimen is not inferior in terms of efficacy, especially among healthy adolescents. However, for vulnerable populations such as individuals with HIV or those who are immunocompromised, a full three-dose schedule remains recommended to ensure adequate protection. We have also added the illustration of evolving dose schedule in section 5. “From 2009 to 2022, the recommended HPV vaccine schedule has continued evolving, reflecting growing evidence from clinical research, real-world data on vaccine effectiveness, and the need to improve coverage and logistical feasibility.” (Line 306-309)
- Why you did not add information related economic purpose? what is the total cost? make a bar graph.
Response: Thank you for your comment! We have added relevant information to the manuscript
(Line 322-328) “The standardized costs per dose are listed as follows:
- Cecolin (bivalent, domestic): 349 RMB per dose
- Cervarix (bivalent, imported): 598 RMB per dose
- Gardasil 4 (quadrivalent, imported): 798 RMB per dose
- Gardasil 9 (nine-valent, imported): 1298 RMB per dose
- Service fee for vaccine administration: 20 RMB per dose”
- Do you have any data regarding long term efficacy and safety of HPV vaccines?
Response: Thank you for your valuable comment! We have included information regarding the long-term efficacy and safety of HPV vaccines into the manuscript. Relevant sections discussing these points have been revised for clarity, highlighting robust evidence from long-term follow-up studies and post-licensure surveillance that support the sustained effectiveness and excellent safety profile of HPV vaccines.
Long-term efficacy:
In section 3: “Clinical trials have demonstrated that the serological response to HPV vaccination far exceeds that of natural infection. Vaccinated women remained seropositive for HPV 16 and18 for up to 113 months, with consistently high levels of IgG antibodies. Notably, IgG levels in vaccinated individuals were 10.8-fold and 10.0-fold higher for HPV 16 and 18, respectively, compared to those generated through natural infection (19). The vaccine offers long-term protection, with the efficacy of the three-dose regimen exceeding 90% for at least 10 years. Ongoing research has suggested continued effectiveness beyond this timeframe (20).” (Line 171-178)
In section 7: “While the immune response from single-dose recipients was lower than that of two- or three-dose recipients, antibody levels remained stable over four years. Over seven years of follow-up, the occurrence of cumulative and persistent HPV16/18 infections in single-dose recipients was similarly low compared to multi-dose recipients, while unvaccinated individuals showed significantly higher infection rates (57).” (Line 445-450)
“Long-term studies, such as the DoRIS study in Tanzania, have observed consistent protection from a single dose for up to 11 years (60)” (Line 453-454)
Safety:
In section 3: “HPV vaccines have consistently demonstrated a strongly safe profile, supported by both clinical trials and real-world post-marketing surveillance data (22). Between 2006 and 2017, over 270 million doses were administered worldwide. Data from the Global Advisory Committee on Vaccine Safety indicate that serious adverse events were extremely rare, with anaphylaxis occurring at a rate of approximately 1.7 cases per million doses and occasional syncope as a frequent, stress-related response rather than a vaccine-specific effect. Other reported minor side effects—such as localized pain, swelling at the injection site, low-grade fever, or mild allergic reactions—were generally mild, self-limiting, and easily managed with proper care (23).” (Line 183-191)
- Title is not clear please modify it.
Response: Thank you for your advice! We revised the title into “Literature Review of HPV Vaccination and Policy Recommendations for a Single-Dose Schedule in China: Opportunities and Challenges”
- The introduction should better explain the rationale and clearly state the knowledge gap this study aims to address.
Response: Thank you so much for your valuable feedback! We agree that a clear articulation of the study’s rationale and knowledge gap is essential. We have revised the introduction section to more explicitly explain the rationale and highlight the existing knowledge gap, together with the novelty of this study.
Rationale: “The focus of this review arises from the WHO’s recent policy update and the ongoing global shift toward reduced-dose HPV vaccination regimens. This topic is particularly timely and relevant for China as the country works to expand HPV vaccine coverage and actively respond to the WHO’s cervical cancer elimination initiative. China is now in a critical transition stage in its HPV vaccination policy, with ongoing adjustments to vaccination schedules, coverage strategies, and public health messaging. The frequent evolution of HPV vaccine policies, coupled with regional epidemiological differences and supply constraints, underscores the rationale for an updated and context-specific synthesis of China’s vaccination framework.” (Line 53-61)
Knowledge gap: “Previous local studies have mainly focused on HPV epidemiology or pricing strategies to improve willingness to pay. However, there remain knowledge gaps regarding China’s recent HPV vaccination practices and the feasibility of adopting a single-dose schedule.” (Line 62-65)
- There are inappropriate uses of periods throughout the manuscript. Please review and revise punctuation.
Response: Thank you for your suggestions! We have double-checked the period punctuation and revised it throughout the text.
16. Verb tenses should be consistent—ideally, use the past tense throughout.
Response: Thank you for your careful review of this manuscript! We have reviewed and revised the verb tenses to ensure consistency in the past tense.
- The manuscript’s grammar requires improvement. Have you considered professional proofreading or English-language editing services?
Response: Thank you very much for your time and constructive comments. We acknowledge the issues raised regarding punctuation, verb tense consistency, grammatical accuracy, and unnecessary repetition. We have applied professional proofreading and carefully revised the manuscript to address these concerns and improved the overall clarity in expression.
Reviewer 2 Report
Comments and Suggestions for Authors
Review of "Evaluating HPV Vaccination in China: A Literature Review and 2 Policy Recommendations for a Single-Dose Schedule"
General comments
First, congratulations on producing a well-written manuscript that sets up the problem with HPV vaccinations in China and describes your narrative review to help address this issue by examining prior literature.
HPV vaccine uptake and completion of full sets of doses are important topics.
You provide a literature review of studies of HPV and its vaccinations in other parts of the world or globally as well as some specific to China. This would be a good background for someone wanting to know more about the academic literature on this topic, such as a student.
You point out many issues that China must overcome to achieve its vaccination goals (e.g., the high price of the HPV vaccine in China compared to the global average), but fail to suggest workable solutions that may assist policymakers in reaching their goals.
The shortfall of over 15 million doses per year is a serious impediment to achieving 90% coverage among 9-14 year old girls. What are your recommendations?
I concur with your recommendation "Tailoring vaccine composition to match regional infection patterns could significantly enhance protective efficacy (ln. 456)
What support do you have for the idea that a single-dose vaccination regime will alleviate barriers discussed in your review? Perhaps you need to add a section following your recommendation (ln 504-510) with prior empirical studies or reports that show specific benefits to overcome the issues that you identified?
Specific comments
Be consistent in your spelling. For example, sometimes it is "Gardasil9" and other times "Gardasil 9"
There is some repetition. For example, the number of doses to cover 90% of 9-14 year-old girls is mentioned a few times. Perhaps you could organize your sections to thoroughly cover these topics all in one place in your review rather than bringing them up multiple times?
Please finish formatting the references at the end of the paper in line with the guidelines for this publication.
Author Response
Thank you very much for taking the time to review this manuscript. We have responded to all the pointers and have revised the manuscript (text in black is your original comments, text in blue is the response and revised contents). We have also updated the reference list. Please find the detailed responses below and the corresponding revisions/corrections highlighted in the re-submitted manuscript.
General comments
First, congratulations on producing a well-written manuscript that sets up the problem with HPV vaccinations in China and describes your narrative review to help address this issue by examining prior literature.
HPV vaccine uptake and completion of full sets of doses are important topics.
You provide a literature review of studies of HPV and its vaccinations in other parts of the world or globally as well as some specific to China. This would be a good background for someone wanting to know more about the academic literature on this topic, such as a student.
You point out many issues that China must overcome to achieve its vaccination goals (e.g., the high price of the HPV vaccine in China compared to the global average), but fail to suggest workable solutions that may assist policymakers in reaching their goals.
The shortfall of over 15 million doses per year is a serious impediment to achieving 90% coverage among 9–14-year-old girls. What are your recommendations?
Response:
Thank you very much for your thoughtful and encouraging comments. We sincerely appreciate the reviewer’s valuable feedback regarding the need for actionable recommendations.
In response to the concern regarding the high vaccine price in China, we propose two strategies: the adoption of single-dose schedule and the implementation of centralized vaccine procurement. The single-dose schedule can significantly reduce out-of-pocket costs for vaccine recipients by lowering both the number of required doses and the associated vaccine and administration fees. For the vaccine price itself, the centralized procurement could increase the purchasing power of the procuring body, which enables the negotiation for lower prices with vaccine manufacturers, thereby making vaccines more affordable. This is a model that has proven effective in other countries and by some literatures, especially when it is coupled with integration into the National Immunization Program (NIP).
To address the shortfall of over 15 million doses per year and the expanded eligibility range (e.g., the recent inclusion of individuals aged 9–45 for the nine-valent HPV vaccine), we propose prioritizing a single-dose vaccination strategy for the targeted girl cohort. This strategy aligns with the WHO’s latest recommendations and would enable the high coverage of HPV vaccines, optimizing the vaccine allocation under resource-restrained situations. Follow-up doses could be reserved for individuals with compromised immune systems or other specific medical indications, maximizing both health equity and cost-effectiveness. Also, we recommend investment in domestic vaccine research and development, together with the implementation of a centralized procurement strategy for imported vaccines, to increase overall vaccine supply.
The discussion section of the manuscript to incorporate these policy-oriented recommendations and provide more practical guidance for decision-makers aiming to improve HPV vaccine coverage in China. We have also added such recommendation in the abstract
“Additional strategies, including increased investment in domestic vaccine research and development, centralized procurement of imported vaccines, and integration of HPV vaccination into the National Immunization Program (NIP), are also critical for policy development. Effective vaccine procurement strategies are essential to ensure sustainable vaccine supply and price (68). Centralized procurement could enhance the government’s purchasing power, facilitating price negotiations with manufacturers and making vaccines more affordable, particularly for countries without Gavi support (69). To address supply shortages, we recommend prioritizing domestic vaccine production alongside centralized procurement of imported vaccines to increase national vaccine availability. Collectively, these strategies offer significant opportunities to strengthen HPV vaccination efforts and improve public health outcomes in China.” (Line 557-568)
I concur with your recommendation "Tailoring vaccine composition to match regional infection patterns could significantly enhance protective efficacy (ln. 456) What support do you have for the idea that a single-dose vaccination regime will alleviate barriers discussed in your review? Perhaps you need to add a section following your recommendation (ln 504-510) with prior empirical studies or reports that show specific benefits to overcome the issues that you identified?
Response:
Thank you for your thoughtful comment. Our intention in raising this point was to emphasize the importance of the nine-valent HPV vaccine which is more aligned with China’s unique infection patterns. For instance, HPV types 52 and 58 are only covered by the imported nine-valent vaccine, which may not be accessible or affordable for Chinese populations. We apologize for the confusion and have reorganized this paragraph.
“From an epidemiological perspective, promoting single-dose vaccination with the nine-valent HPV vaccine may offer greater protective benefits in China due to its distinct HPV genotype distribution. Epidemiological data from China reveal a unique HPV genotype distribution, with HPV 52, 58, and 18 showing high prevalence rates (11). While bivalent and quadrivalent vaccines primarily target HPV 16 and 18, they may inadequately address the broader range of high-risk genotypes circulating in China. This mismatch potentially limits their population-level protective impact. As single-dose regimens have demonstrated comparable efficacy and duration of protection to multi-dose regimens, prioritizing the nine-valent vaccine in a single-dose schedule could enhance coverage of locally prevalent high-risk HPV types. This approach would broaden the spectrum of protection, offering a more epidemiologically relevant and equitable solution for cervical cancer prevention in China.” (Line 492-503)
Specific comments
Be consistent in your spelling. For example, sometimes it is "Gardasil9" and other times "Gardasil 9". There is some repetition. For example, the number of doses to cover 90% of 9-14 year-old girls is mentioned a few times. Perhaps you could organize your sections to thoroughly cover these topics all in one place in your review rather than bringing them up multiple times? Please finish formatting the references at the end of the paper in line with the guidelines for this publication.
Response:
Thank you for your careful reading and constructive feedback. We appreciate the reviewer’s attention to the details. In response, we have carefully reviewed the manuscript to ensure consistency in terminology and spelling throughout the text
In the revised version, we standardized the use of “Human Papillomavirus” by spelling it out only at its first mention (Line 40) and subsequently using the abbreviation “HPV.” When referring to vaccine types, we use the format “HPV [variant number]” (e.g., HPV 16, HPV 18). When listing several HPV variants at the same time, we mention “HPV” only once, followed by the additional type numbers (e.g., HPV 18, 52, and 58) to avoid unnecessary repetition. We standardized vaccine names, using “Gardasil 9” to refer to the nine-valent vaccine and “Gardasil 4” for one of the two quadrivalent vaccines, in line with the official product names used by their manufacturer—Merck Sharp & Dohme.
We also acknowledge the noted repetition of certain information. To improve clarity and cohesion, we have removed some unnecessary repetition and reorganized the relevant information to reduce redundancy of the review.
Reviewer 3 Report
Comments and Suggestions for Authors„Evaluating HPV Vaccination in China: A Literature Review and 2 Policy Recommendations for a Single-Dose Schedule”
The authors presented an interesting and very current topic. I believe that the review article is well written. My dissatisfaction consists of:
1) the lack of indication of which diseases the HPV virus is responsible for (the authors write only about cervical cancer, and there are more possible cancers)
2) the authors do not indicate that cancers caused by HPV also affect boys and men
3) the authors should also indicate the role of men as a reservoir of the virus, and consequently the need to vaccinate the male part of the population - such strategies are undertaken by many countries (please indicate this in the article) and this is a very important aspect from the point of view of epidemiology
Thank you.
Author Response
Thank you very much for taking the time to review this manuscript. We have responded to all the pointers and have revised the manuscript (text in black is your original comments, text in blue is the response and revised contents). We have also updated the reference list. Please find the detailed responses below and the corresponding revisions/corrections highlighted in the re-submitted manuscript.
The authors presented an interesting and very current topic. I believe that the review article is well written. My dissatisfaction consists of:
1) the lack of indication of which diseases the HPV virus is responsible for (the authors write only about cervical cancer, and there are more possible cancers)
Response: Thank you very much for this insightful comment. While we did briefly mention the broader disease burden of HPV in the Virology and Epidemiology section:
“Persistent HPV infection, especially with high-risk types, is responsible for over 95% of cervical cancer cases as well as many HPV-related diseases.” (Line 41-42)
“While low-risk types are occasionally linked to neoplastic changes, high-risk types, particularly HPV 16 and 18, are frequently observed in cancers of the cervix (CIN), vulva (VIN), vagina (VaIN), penis (PIN), and anus (AIN) (10). Most HPV infections are transient and resolve spontaneously within 1–5 years. However, persistent infection with high-risk types can lead to the development of precancerous lesions and invasive cancers.” (Line 105-109)
We agree that this point deserves a clearer indication. We have revised the manuscript, using more accurate terms such as HPV-related diseases”, not limited to cervical cancer when it comes to the benefits of HPV vaccination programs. For instance, “The WHO emphasizes the safety, cost-effectiveness, and efficacy of HPV vaccines in reducing HPV-related infections, not only the cervical cancer cases, but also other associated cancers.” (Line 152-154)
“While HPV infection is most discussed with cervical cancer, this group of viruses has broader oncogenic impact and is implicated in multiple other malignancies. Persistent HPV infection is responsible for over 95% of cervical cancer cases, and high-risk HPV types also significantly contribute to vulval, vaginal, penile, anal, and head-and-neck cancers (1).” (Line 112-116)
2) the authors do not indicate that cancers caused by HPV also affect boys and men
Response: Thank you for highlighting this important point. We acknowledge that while cervical cancer specifically affects the female population and is one of most discussed HPV-related diseases. Other HPV-associated cancers—including penile, anal, oropharyngeal, and head-and-neck cancers—can also affect men. Moreover, men are also part of HPV transmission and are important targets for vaccination, both for their own protection and to reduce transmission. Although the primary public health goal of HPV vaccination remains the prevention of cervical cancer in girls, it is also essential to recognize the broader impact of HPV infection. Including boys in vaccination strategies provides direct protection and contributes to reducing overall HPV transmission within the population.
We have revised the manuscript to emphasize that HPV-related diseases are not exclusive to females and emphasized this in future studies.
“HPV infection is one the most common sexually transmitted disease, affecting individuals of all genders. While extensive research has focused on HPV-related diseases in women, particularly cervical cancer, male populations have received comparatively little attention due to higher burden of HPV-related morbidity and mortality observed in women (12). The highest rates of genital HPV infection are observed in young women under 25 years old, with a global prevalence of approximately 30%. A secondary peak in HPV prevalence commonly occurs in women aged 50–60 years, though regional variations exist (13). Despite cervical cancer being a female-specific disease, HPV infection itself is not gender-restricted. Males are also susceptible to HPV-related disease burdens and serve as vectors in the transmission of the virus (12).” (122-131)
“Third, this review focuses primarily on cervical cancer prevention in females, overlooking the broader public health impact of HPV infection. While HPV is a significant health issue affecting all genders, current prevention efforts remain largely centered on cervical cancer, with limited attention given to its role in other HPV-related cancers and diseases (70). The role of male vaccination is also underexplored, even though including boys in HPV vaccination strategies could enhance herd immunity, reduce virus transmission, and protect against HPV-related diseases in all genders (71).” (Line 615-622)
3) the authors should also indicate the role of men as a reservoir of the virus, and consequently the need to vaccinate the male part of the population - such strategies are undertaken by many countries (please indicate this in the article) and this is a very important aspect from the point of view of epidemiology
Response: We appreciate this thoughtful observation. Indeed, men play a significant role in the HPV epidemiology, and vaccinating boys and men contributes to herd immunity and overall disease burden reduction. Several countries, including the United States, Australia, and parts of Europe, have implemented gender-neutral HPV vaccination strategies. In China, vaccinating girls remains the main target, male HPV vaccination is not widely recommended yet.
We agree this is an essential consideration from an epidemiological perspective. We have revised the manuscript, using a more gender-neutral expression in section 2 Virology and Epidemiology of HPV to describe the overall vaccination program. “As such, HPV vaccination for both genders is the cornerstone of primary prevention, provides direct protection but also enhances herd immunity and resilience to disruption of cancer prevention (15).” (Line 150-152)
We included the impact of gender-neutral vaccination on vaccine demand and supply and added this discussion in section 7.
“Moreover, the lack of routine male HPV vaccination further compounds the vaccine supply shortage. Adopting a gender-neutral HPV vaccination strategy could reduce population-level transmission, combat misinformation, minimize vaccine-related stigma, and promote gender equity (66). Globally, only 43 countries and 4 territories have implemented gender-neutral HPV vaccination programs, which are predominantly located in high-income and upper-middle-income region (67). In contrast, China’s vaccination efforts remain largely focused on girls. However, gender-neutral HPV vaccination plays a critical role in controlling transmission and contributes to herd immunity, as an effective strategy to improve the resilience to disruption of cancer prevention and overall disease burden reduction (15). If vaccination targets in China are further expanded to include males, vaccine demand will increase substantially. Additionally, the evolving vaccination guidelines, though reflective of efforts to align with WHO recommendations, can create confusion among healthcare providers as well as the public. Frequent updates to dose schedules and eligibility criteria, particularly during this transitional phase, may hinder consistent implementation and compromise the clarity and effectiveness of vaccination efforts.” (Line 537-552)
We also included the male vaccination in the challenges & opportunities section, emphasizing the benefit of male vaccination and the even more intense demand in HPV vaccines. We also describe the dilemma in limitation and future research, calling for more research into HPV-related diseases in males and vaccine recommendation.
“Third, this review focuses primarily on cervical cancer prevention in females, overlooking the broader public health impact of HPV infection. While HPV is a significant health issue affecting all genders, current prevention efforts remain largely centered on cervical cancer, with limited attention given to its role in other HPV-related cancers and diseases (70). The role of male vaccination is also underexplored, even though including boys in HPV vaccination strategies could enhance herd immunity, reduce virus transmission, and protect against HPV-related diseases in all genders (71).” (Line 615-622)
“Further evaluation of school-based delivery, public education strategies, and gender-neutral vaccination could support broader and more equitable vaccine access. Re-search exploring the prevention of non-cervical HPV-related diseases and the benefits of male vaccination will be critical for informing a more comprehensive, inclusive HPV control strategy. Together, these efforts can strengthen China’s contribution to global cervical cancer elimination and HPV-related disease prevention.” (Line 628-633)
Round 2
Reviewer 1 Report
Comments and Suggestions for Authors
Where is the certificate for English editing ?
where is the list of abbreviation ?
I think you just copy figure 2 ? why you did not prepare by yourself ?
Author Response
Thank you very much for taking the time to review this manuscript. We have responded to all the pointers and have revised the manuscript (text in black is your original comments, text in blue is the response and revised contents).
- Where is the certificate for English editing?
My response: Thank you for your feedback and for pointing out concerns about the English and period spacing in the manuscript. To ensure the language quality, I used Grammarly, a professional writing assistant tool, to edit the English and correct spacing issues. I have carefully reviewed the text again to make sure no errors remain. Please let me know if any specific areas require further improvement.
- where is the list of abbreviation?
My response: Thank you for your suggestion. We have prepared the list of abbreviations in the manuscript for your review.
- I think you just copy figure 2. why you did not prepare by yourself?
My response: Thank you for your feedback. I would like to clarify that Figure 2 was entirely and originally created by me using Canva, an online design platform. To verify this, you can visit the link provided below, which clearly displays my name and email address as the creator and owner of the figure. I confirm that this figure is completely original and not sourced or copied from any external material. Furthermore, for additional verification, we have zipped and submitted the editable PowerPoint (.pptx) version of the figure in the Figures, Graphics, Images tab.
Link to Canva site:
https://www.canva.cn/design/DAGpUlDWvDo/kKLxbdyqOUsGBwWtcw419Q/edit?utm_content=DAGpUlDWvDo&utm_campaign=designshare&utm_medium=link2&utm_source=sharebutton